# Family Variables and Quality of Life in Children with Down Syndrome: A Scoping Review

**DOI:** 10.3390/ijerph18020419

**Published:** 2021-01-07

**Authors:** Anna Lee, Kathleen Knafl, Marcia Van Riper

**Affiliations:** 1School of Nursing, Korea University, Seoul 02841, Korea; 2School of Nursing, University of North Carolina at Chapel Hill, Chapel Hill, NC 27599-7460, USA; kknafl@email.unc.edu (K.K.); vanriper@email.unc.edu (M.V.R.)

**Keywords:** down syndrome, children, adolescents, quality of life, family variables

## Abstract

The purpose of this scoping review was to identify the family and child quality of life variables that have been studied in relation to one another in children with Down syndrome, the frequency with which different relationships have been studied, and the extent to which family variables were the focus of the research aims. A literature search was conducted to find studies published between January 2007 and June 2018. The initial search yielded 2314 studies; of these, 43 were selected for a final review. Researchers most often addressed family resources and family problem-solving and coping concerning child personal development and physical well-being. Little attention to child emotional well-being was observed, with none considering family appraisal of child emotional well-being. The relationship between family variables and child QoL rarely was the primary focus of the study. Methodologically, most reviewed studies used cross-sectional designs, were conducted in North America and based on maternal report. From future research considering the issues found in this review, healthcare providers can obtain an in-depth understanding of relationships between children and family variables.

## 1. Introduction

Down syndrome (DS) is the most frequently occurring chromosomal disorder, with an incidence of 1 in 1000 to 1 in 1100 live births worldwide [1]. With improved medical and surgical care, the survival of children born with DS has significantly improved over recent decades [2]. In the United States of America (USA), the 20-year survival probability for individuals born with DS is 88% [2]. Although mortality rates during the first few years of life continue to be higher for children with DS than for the general population, the average life span for individuals with DS has increased from 25 years in 1983 to over 60 years at present [3,4]. This dramatic increase in life expectancy for individuals with DS has prompted both service providers and researchers to pay greater attention to quality of life (QoL) in individuals with DS [5,6]. The investigation of QoL is essential for evaluating personal outcomes and guiding organization- and system-level policies and practices that aim at improving the lives of people with DS [7]. An in-depth understanding of QoL in children and adolescents with DS as well as its relationship to associative factors that weaken or strengthen children’s QoL will contribute to healthcare providers’ ability to better address the needs of the population.

Schalock et al. [7] conceptualized QoL for persons with intellectual disabilities (ID) as a multidimensional phenomenon composed of the following eight core domains: emotional well-being, physical well-being, material well-being, personal development, self-determination, interpersonal relations, social inclusion, and rights. These eight core domains, along with the three most commonly used indicators for each domain, are listed in Table 1 [7]. The framework developers additionally grouped these domains into three higher-order factors: well-being (emotional, physical, and material), independence (personal development and self-determination), and social participation (interpersonal relations, social inclusion, and rights).

Children with DS show lower levels of the overall QoL than children without DS though [9], with varying levels across the QoL domains. Children with DS are more likely to show high levels of emotional well-being such as a positive self-concept [10]. However, lower levels of physical well-being in this population are reported compared to typically developing (TD) children [11]. Investigators have reported that children with DS show fewer maladaptive behaviors and better social competence than children with other IDs [12,13]. Children with DS, however, have been shown to have more problem behaviors and poorer social capabilities than TD counterparts [14]. Investigators have also reported that individuals with DS perceived their relationships with family members as positive and supportive [10]. Yet, studies of peer relationships of children and adolescents with DS revealed difficulties in forming and sustaining friendships [15,16].

Findings from existing studies provide a somewhat mixed picture of QoL in individuals with DS. This is due, in part, to the fact that there has been a great deal of variability in terms of how QoL is defined and measured [17,18]. In addition, individuals with DS have often been considered a homogeneous group; researchers have paid limited attention to how innate individual traits (e.g., sex, age) may influence QoL. Moreover, researchers interested in identifying factors that may enhance or diminish QoL in individuals with DS have examined a wide variety of factors (e.g., family variables, community factors), making it difficult to compare findings across studies. Because individuals with DS are living longer, it is vital to identify which factors enhance or diminish their QoL as an important step in developing interventions that aim at strengthening modifiable factors known to enhance QoL.

Based on prior research, there is strong evidence that family variables are especially important influences on the QoL of children with DS, e.g., [19,20]. The family variables addressed in previous studies are consistent with key concepts of the Resiliency Model of Family Stress, Adjustment and Adaptation [21], which conceptualizes core family characteristics that contribute to adaptation outcomes for family members and the family system. The concepts of this model include family demands, family appraisal, family problem-solving and coping, family resources, and family adaptation. The definitions for these family variables are included in Table 2. Family adaptation refers to the outcome of families’ efforts to attain a new balance between the needs of individual family member(s) and those of the family as a whole when the family faces a crisis situation [21]; the optimum levels of well-being of individual family members as well as a family unit indicate successful family adaptation.

Based on Schalock and colleagues’ [18] conceptualization of QoL in individuals with ID and McCubbin and colleagues’ [21] Resiliency Model of Family Stress, Adjustment and Adaptation, the intent of this scoping review was to (1) identify the family and child QoL variables that have been studied in relation to one another in children with DS, (2) determine the frequency with which different relationships have been studied, and (3) examine the degree to which family variables were the focus of the research aims.

## 2. Materials and Methods

Consistent with aims, we undertook a scoping review to examine the range of family and child QoL relationships that had been studied based on the steps delineated by Peters et al. [22] for conducting scoping reviews. We were interested in identifying both gaps in the literature and areas where there was a sufficient body of research for undertaking future systematic reviews. A formal assessment of the quality of the included studies is not carried out because scoping reviews seek to provide an overview of all reviewed studies regardless of quality [22,23].

### 2.1. Inclusion and Exclusion Criteria

To be included in the current sample, a report had to: (a) address the relationship between family variables and QoL in individuals with DS aged 0–21 years old and (b) have been published in an English language peer-reviewed journal. According to the United Nations Convention on the Rights of the Child, children are persons up to the age of 18 [24]. Considering generally identified developmental delays among individuals with DS, we included the sample aged up to the age of 21. When the age range for a sample exceeded 21 years, the study was included only if the sample’s mean age fell within the eligibility criteria. This review also included reports with multiple conditions as long as the variables related to DS could be identified. Exclusion criteria for this review were (a) the report did not address the relationships between family variables and children’s QoL, or (b) the report addressed methodological issues, instrument development, or practice guidelines. Review papers also were excluded.

### 2.2. Searching

A literature search in PubMed, CINAHL, Web of Science, and PsycINFO was conducted to find studies published between January 2007 and June 2018. Given changes in the life expectancy of children with DS, we wanted to focus on relatively recent studies. At the time the search was conducted, we were targeting studies completed in the prior 10 years. We consulted with a research librarian who had expertise in searching electronic databases to identify keywords and medical subject heading (MeSH) terms. We employed several combinations of keywords and MeSH search terms in each electronic search engine (Table A1). The first group of search terms consisted of synonyms for Down syndrome. The second group of search terms included quality of life or well-being. The last group of search terms related to family variables.

Two authors (A.L., K.K.) independently screened article abstracts to identify potentially relevant articles. Full texts of these articles were obtained, and the authors independently reviewed the texts. The authors discussed disagreements to reach a consensus on the final sample. The Covidence platform (www.covidence.org), an online tool specifically designed to enhance the efficiency and thoroughness of article screening and review, was used to screen reports and identify disagreements.

### 2.3. Extracting Results

After obtaining a pdf of each report included in the sample, the first author used a structured template to extract information on the characteristics of each study, including the study’s purpose, design, participants, and the children’s QoL and family variables that were measured. The second author reviewed the extraction for accuracy and completeness. Following a discussion to reach a consensus, the first author made any needed revisions to the extractions.

### 2.4. Analysis

Family variables drawn from reviewed studies were grouped into the four family variables: family demands, family appraisal, family resources, and family problem-solving and coping of the Resiliency Model [21]. Schalock et al.’s [18] eight domains (emotional, physical, material well-beings, personal development, self-determination, interpersonal relation, social inclusion, and rights) were used to categorize variables that have been used to assess QoL in children with DS.

To address our third aim, examining the degree to which family variables were the focus of the research aims, we adapted criteria developed by Knafl and colleagues [25] to categorize study aims as “fully”, “partially” or “minimally” focused on family–child QoL relationships. Our adaptations entailed incorporating our conceptualizations of family and child QoL into our definition of the three categories. As a check on the categorization process, the first two authors independently categorized each study. The third author identified discrepancies and conferred with the first two authors to reach consensus on how studies were categorized.

## 3. Results

In the following sections, we provide an overview of the studies included in the sample (Table 3). Following that, we report the family variables linked to QoL for each study in the sample. We present the specific family variables categorized by the Resiliency Model [21] and the children’s QoL variables categorized by the QoL conceptualization [18] with the associated measures used in the studies available in Table A2. With the exception of Scott et al. [26] who measured children’s overall well-being, the QoL variables reported in all the reviewed studies addressed the domains in Schalock et al.’s [18] conceptual framework. In Table 4, the categorization regarding family focuses in stated aims of reviewed studies as well as the categorization scheme are described. In Table 5, we present a summary of the relationships between the family variables and children’s QoL.

### 3.1. Description of Research Reports

The electronic search yielded a total of 2314 studies (Figure 1). Of these, 43 met the eligibility criteria and were selected for a final review. The most common study design was quantitative cross-sectional (*n* = 27, 63%). Eight were quantitative longitudinal studies, whereas six were qualitative cross-sectional studies. There was one randomized controlled trial study and one cross-sectional mixed methods study.

Twelve studies were conducted in the USA (28%), eight in Italy (19%), and five in Australia (12%). Two studies were identified with sample drawn from Canada, Turkey, and the United Kingdom and one study was identified with a sample from Brazil, China, Iran, Ireland, Korea, Netherlands, New Zealand, Norway, Saudi Arabia, Spain, Taiwan, Thailand, and Yemen. Most were conducted in North America or Europe, not highlighting Africa. Across the 43 studies included in this review, data were collected about 4207 children and 4258 family members.

Twenty-three studies indicated that the data came from parents or caregivers [27,28,29,30,31,32,33,34,35,36,37,38,39,40,41,42,43,44,45,46,47,48,49]. Authors of 16 studies reported that the mother was the sole respondent for all data [50,51,52,53,54,55,56,57,58,59,60,61,62,63,64,65]. In one study [66], fathers were the only respondents. Only two studies gathered data directly from the person with DS [26,67]. In one study [68], data came from an existing data set. The mean age of children ranged from 19 months to 21 years and the mean percentage of male children was 59.2 (Table 3).

### 3.2. Family Variables and Children’s QoL

#### 3.2.1. Family Demands and Children’s QoL

Family Demands and Children’s Well-Being. Scott et al. [26] studied the relationship between barriers to social participation (family demands) among individuals with DS, and their perceptions of a good life (overall well-being) and what made them happy or sad (emotional well-being), and types of residence (material well-being).

In seven studies, authors addressed the relationship between family demands and children’s physical well-being. Four studies addressed parents’ perceptions of obstacles within family/community contexts that hindered physical activities (PAs) of children with [28,30,38,50]. Three studies examined the relationship between parental psychological distress and children’s health, specifically, between parental depression and health problems [33], parental concerns and sedentary behavior [40], and parental concerns and children’s body mass index (BMI) [44].

Family Demands and Children’s Independence. Seven studies explored the relationship between family demands and children’s personal development. Five studies focused on the relationship between family demands and children’s behavior and particularly examined how individual family members’ psychological distress was related to children’s behavioral issues [32,52,59,60,62]. The association between family psychological distress and children’s social/cognitive functioning was investigated in four studies [32,33,60,62]. One study evaluated parental neuroticism/stress and its relationship to the children’s educational achievement [46]. Burke et al. [32] measured the relationship between parental depression and children’s motivational strengths such as independence and morality (self-determination).

Family Demands and Children’s Social Participation. Lyons et al. [42] explored which barriers influenced activities and participation in everyday activities among children with DS (social inclusion). Choi and Yoo [33] examined how parental depression and family strain were related to stigma/discrimination experiences of children with DS (rights).

#### 3.2.2. Family Appraisal and Children’s QoL

Family Appraisal and Children’s Well-Being. Izquierdo-Gomez et al. [40] studied the relationship between parents’ perceptions of the importance of PA and children’s sedentary behaviors (physical well-being).

Family Appraisal and Children’s Independence. Authors of two studies examined the relationship between family appraisal variables and children’s independence. Minnes et al. [59] reported on the relationship between positive parental views of raising a child with DS or DD and the child’s adaptive/maladaptive behaviors (personal development).

Burke et al. [32] addressed the family appraisal variables of parental perceptions on children’s positive contributions to family life, the rewards from having a child with DS, and the rewards/worries as their child transitioned to adulthood in relation to the child variables from the personal development domain (e.g., children’s intelligence quotient [IQ], maladaptive behaviors, personalities) and the self-determination domain (e.g., motivational strengths and styles).

#### 3.2.3. Family Resources and Children’s QoL

Family Resources and Children’s Well-Being. In a study that gathered data directly from individuals with DS, Scott et al. [26] examined the relationships among facilitators (family resources) of social participation for individuals with DS, their perceptions of a good life (overall well-being), and what made them happy or sad (emotional well-being) as well as the type of residence (material well-being). Howell et al. [56] examined family climate as a predictor (family resources) of loneliness among children with DS (emotional well-being).

Twelve studies examined the relationship between family resources and children’s physical well-being. Authors of five studies assessed family resources including parental education (family resources from individual family members), family functioning (family resources from the family as a unit), or community service (family resources from the community) and their associations with children’s health issues (e.g., general health problems, oral health) [29,33,44,47,68]. Four studies explored family and community supports contributing to PA in children with DS [28,30,38,50]. Three studies investigated the relationship between family resources (individuals, family, and community) and children’s leisure activities [37,40,43].

Family Resources and Children’s Independence. The most frequently studied relationship between family and child variables was between family resources and children’s personal development (*n* = 20, 47%), with most of these examining the relationship between individual family resources and children’s cognitive functioning [33,39,48,53,56,58,62,63,66]. For example, Evans and Uljarević [39] investigated the possible influence of parental education on children’s IQ. The relationship between individual family member resources and children’s social competence was addressed in seven studies [33,41,47,57,58,60,66]. For example, Karaaslan, and Mahoney [57] examined the relationship between parental interactive styles (e.g., responsivity, sensitivity) and social engagement in children with DS.

Other investigators addressed different aspects of the relationships between one or more of the three types of family resources (individual, family, and community) and children’s personal development, with resources related to the family unit studied most frequently. Resource variables were examined in relation to children’s personal development, including behavioral issues [56,59,60,61,62,67] and language functioning [35,45,54].

The relationship between family resources and self-determination was addressed in the study by Wang et al. [47] who evaluated the relationships between parental education and family income and children’s self-management ability (e.g., doing something independently, planning).

Family Resources and Children’s Social Participation. Oates et al. [43] examined family resources including the availability of parental time, family income, and access to transportation and their relationship to friendship interactions among children with DS (interpersonal relations). Lyons et al. [42] explored the effects of family roles on participation in everyday activities among children with DS (social inclusion). Choi and Yoo [33] reported on the relationship between family cohesion and stigma/discrimination experiences of children with DS (rights).

#### 3.2.4. Family Problem-Solving and Coping and Children’s QoL

Family Problem-Solving and Coping and Children’s Well-Being. Three studies addressed the relationship between family problem-solving and coping and children’s physical well-being. Two studies respectively assessed relationships between the pattern of family communication and children’s health problems [33] as well as between family leisure activities and children’s sedentary behaviors [40]. Polfuss et al. [44] examined the relationship between parental feeding behaviors and children’s BMI.

Family Problem-Solving and Coping and Children’s Independence. In this review, the second most studied relationship was between family problem-solving/coping and children’s personal development (*n* = 14, 33%). Among them, five studies examined the relationships between children’s language development and parental variables such as maternal functional language [65], strategies to enhance children’s communication skills [27,31,49] and parents’ translation of children’s symbolic communicative gestures [36]. Four studies assessed the relationship between parental responses toward children’s play and the quality of children’s play behaviors [34,51,64,66]. Three studies measured the relationships between parental coping styles and children’s behavioral issues [32,59] or educational achievement [46]. Gilmore et al. [55] examined how maternal verbal strategies were associated with children’s persistence in task completion. Choi and Yoo [33] studied how family communication patterns were related to children’s developmental levels and sociability. Burke et al. [32] assessed how parental coping styles were associated with children’s motivational strengths and styles (self-determination).

Family Problem-Solving and Coping and Children’s Social Participation. Only one study addressed this relationship. In Choi and Yoo’s [33] study, the relationship between the pattern of family communication and children’s stigma/discrimination experiences, a variable associated with children’s rights, was addressed.

### 3.3. Degree of Family Focus

As summarized in Table 4, sixty percent (*n* = 26) of the studies were classified as full focus studies, indicating that all study aims addressed the relationship between one or more of the variables encompassed by the conceptual models underpinning this review. For example, Burke and colleague’s [32] study examined the relationships between parental depression (family demands), parental coping styles (family problem-solving and coping), parents’ perceptions on their children’s transition to adulthood (family appraisal) and the children’s behavioral characteristics, and Niccols and colleagues [61] explored the relationship between aggression in children with DS and maternal sensitivity (family resources). Seventeen of these studies addressed the relationship between child QoL and family resources as measured by a characteristic of an individual family member (e.g., maternal responsiveness). Only nine of the 26 studies categorized as fully family focused included a family system measures (e.g., a measure for family climate or cohesion) [33,35,43,45,46,56,59,60,67]. Although our categorization was consistent with the Resiliency Model, the results highlight researchers’ predominant focus on individual family members rather than the family system as a whole. Additionally, we found that eight of the nine studies measuring a family system variable reported significant relationships between family variables and child QoL variables. For example, Howell et al. [56] and Minnes et al. [59] studied family climate and family empowerment, respectively. The investigators found a significant relationship between these family variables and children’s loneliness [56] and maladaptive behaviors [59]. The only study in which a non-significant relationship was reported addressed the relationships between family social support and children’s adaptive/problem behaviors [60].

Ten studies were categorized as partially family focused because only some of the study aims addressed the relationship between family and child QoL variables. For example, Al-Sufyani and colleagues [29] examined the relationship between the children’s oral health and both clinical variables and parental characteristics. Of these ten studies, only four studies addressed a family system variable [40,42,47,48], with the remaining addressing individual family members’ characteristics such as maternal stress or educational level. Seven studies were categorized as minimally family focused because they measured a Resiliency Model variable, but did not address the relationship between family factors and child QoL in the study aims. While the family variables of the studies in the partial focus category reflected all family variables of the Resiliency Model, those under the minimal focus category corresponded to only two family variables (family demands and family resources).

## 4. Discussion

Drawing from a sample of research reports published in the past decade, this scoping review of 43 peer-reviewed publications identified the family variables that investigators have studied in relation to children’s QoL and the extent to which specific relationships were examined. Notably, a strength of the review was its conceptual grounding in two well-established conceptual frameworks: The Resiliency Model of Family Stress, Adjustment and Adaptation [21] and Schalock et al.’s [18] QoL conceptualization. These frameworks provided a useful structure for differentiating broad categories family and child variables that contributed to the identification of the extent to which core dimensions of QoL have been examined in relation to family variables.

The reports published between 2007 and 2018 examined relationships between family variables and children’s QoL across a broad spectrum of both family and child variables. All the core family variables reflected in the conceptual framework (family demands, family appraisal, family resources, and family problem-solving/coping) were studied in relation to one or more children’s QoL variables in this review.

Investigators have primarily focused on family variables addressing family resources (*n* = 20) and family problem-solving and coping (*n* = 14) and their relationships to QoL variables regarding the personal development of children with DS. Investigators have also examined the relationship between family resources and children’s physical well-being (*n* = 12).

Although somewhat less frequently studied, investigators have also focused attention on the relationships between family demands and children’s physical well-being (*n* = 7) as well as those between family demands and children’s personal development (*n* = 7); these studies, however, are relatively few compared to the number of studies addressing the relationship of family resources and problem-solving and coping to children’s QoL.

Other relationships between family and child variables were addressed in only 1–3 studies (e.g., family demands and children’s emotional well-being). No studies included in this review addressed the following relationships: family demands and children’s interpersonal relations; family appraisal and children’s overall well-being, emotional well-being, material well-being, interpersonal relations, social inclusion, and rights; and family problem-solving and coping and children’s overall well-being, emotional well-being, material well-being, interpersonal relations, and social inclusion.

Taken together, the present review shows a trend in concentrating on family resource variables that were used to examine their relations to children’s QoL. Since investigators often collected demographic data on family income and parents’ educational level, e.g., [35,44,54,68], which we categorized as a family resource it is not surprising that this was a frequently studied relationship. Although frequently studied, demographic variables are not typically targeted for interventions. On the other hand, family appraisal variables reflecting family members’ views of the stressors and demands they confront and their ability to address them [21] were rarely studied despite being a potentially modifiable aspect of family life.

Researchers examining the personal development of children with DS most often studied the children’s competence related to cognitive, language, and social functioning, all of which play a pivotal role in children’s QoL. Children with DS have been shown to have deficits in verbal processing in addition to more behavioral and social problems than their typically developing (TD) counterparts [69,70], and it is understandable that researchers have focused attention on the competence of children with DS. Research is needed on potentially modifiable factors contributing to competence and other aspects of personal development.

Since children with DS are at higher risk for having co-occurring health problems such as heart disease, leukemia, and obesity [71,72], it is not surprising that researchers are addressing physical well-being, most often as it relates to family demand or family resources.

In contrast, there was minimal attention given to emotional well-being, with only three studies addressing the issue and none considering the family appraisal of children’s emotional well-being [32,40,59]. This is a notable gap in the literature, and research is needed to determine if and how parents assess and support the emotional well-being of children with DS.

There was also limited research on the relationship between family variables and children’s material well-being, self-determination, interpersonal relations, social inclusion, and rights, despite these being core components of children’s QoL. Particularly, as children with DS age, the self-determination domain that is pertinent to independence and autonomy, as well as the social inclusion domain, have been shown to be increasingly important components of QoL [73,74]. An in-depth understanding of all the QoL domains in relation to family variables is needed to support families in their efforts to enhance and sustain QoL for children with DS. Since children with DS often live in the family home well into adulthood [75,76], attention also needs to be given to how the relationships between children’s QoL and family variables change over time.

Over 80% of reviewed studies were categorized as fully or partially family focused (*n* = 36) because study aims included Resiliency Model variables. However, the studies mostly focused on individual family members (*n* = 23) (e.g., parental depression). Relatively few studies included family system measures such as family flexibility or family cohesion. The relationships between family system variables such as adaptation and functioning (e.g., family climate, family function), and child QoL variables were predominately significant, which provides evidence of the contribution of family system variables to child’s QoL. Future research is warranted to include more family system measures to gain an in-depth understanding of a family as a whole.

Although our analysis assessed the degree of family focus by examining the variables reflected in the investigators’ statement of aim(s), it should be noted that relatively few authors framed their aims in terms of addressing the relationship between family variables and child QoL. Using the Resiliency Model and Schalock’s conceptualization of children’s QoL, we were able to identify the ways in which the relationships between family and child QoL variables were addressed in these studies, thereby highlighting an important aspect of this body of literature.

In terms of methodological issues drawn from the current scoping review, most of the included studies used cross-sectional designs and maternal report as the primary data source. Since children with DS tend to live in the family home longer than other children do, it is especially important to have longitudinal studies that address the family predictors of QoL in children with DS over time. Longitudinal studies would allow investigators to examine the changing relationship between family and child QoL variables over time.

It has been frequently noted as a critical methodological issue that studies have relied heavily on maternal reports of data about the family system, possibly leading to a skewed view of child and family life [77,78]. Approximately half of the included studies reported that their data came solely or predominantly from mothers (*n* = 21, 49%). Data from children with DS and other family members, especially fathers and siblings, will provide a more comprehensive view of the relationship between children’s QoL and family variables, including different family members’ distinct contributions to children’s QoL. In addition to including children and multiple family members in the study sample, investigators are also encouraged to incorporate dyadic or family systems analyses.

We found that most included studies were conducted in North America or Europe. Children with DS and their families in other parts of the world could have different experiences based on culturally grounded beliefs about intellectual disabilities such as DS [79,80]. In some cultures, disability is accompanied by a sense of shame and stigma [79,80]. Additionally, societal expectations about the family’s responsibility to care for these children, and social services and supports could vary across countries [79,80]. It would be interesting to examine the extent to which relationships of children with DS and their families are universal or culturally bound through studies including a wider array of families.

The current review did not entail a quality assessment. Yet, it is consistent with accepted scoping review methods and the purpose of this review was to map the nature of the literature regarding family variables that have been studied in relation to QoL in children with DS. Since this scoping review included studies published between 2007 and 2018, we might have missed other relevant articles. To minimize the risk of missing relevant articles, we worked with a librarian who had expertise in searching electronic databases. Finally, we only included studies written in English; possibly relevant studies written in different languages were excluded.

## 5. Conclusions

This review identified the relationships between family and child variables that investigators have most frequently discussed in recent years. Our results both highlight gaps in the current body of research on the relationships between family variables and QoL in children with DS, and promising areas for conducting integrative or systematic reviews of study results. Based on the current findings, conducting systematic reviews including analyses of statistical significance will be salient. Particularly, family resources and their contribution to the personal development of children with DS could be targeted for systematic reviews in that the current review found evidence of a sufficient number of relevant studies. More longitudinal studies are also needed, including studies with multiple family members and culturally diverse samples as well as family system measures considering the family as a unit. Future research should take into consideration the issues found in the current scoping review for children with DS and their families.

## Figures and Tables

**Figure 1 ijerph-18-00419-f001:**
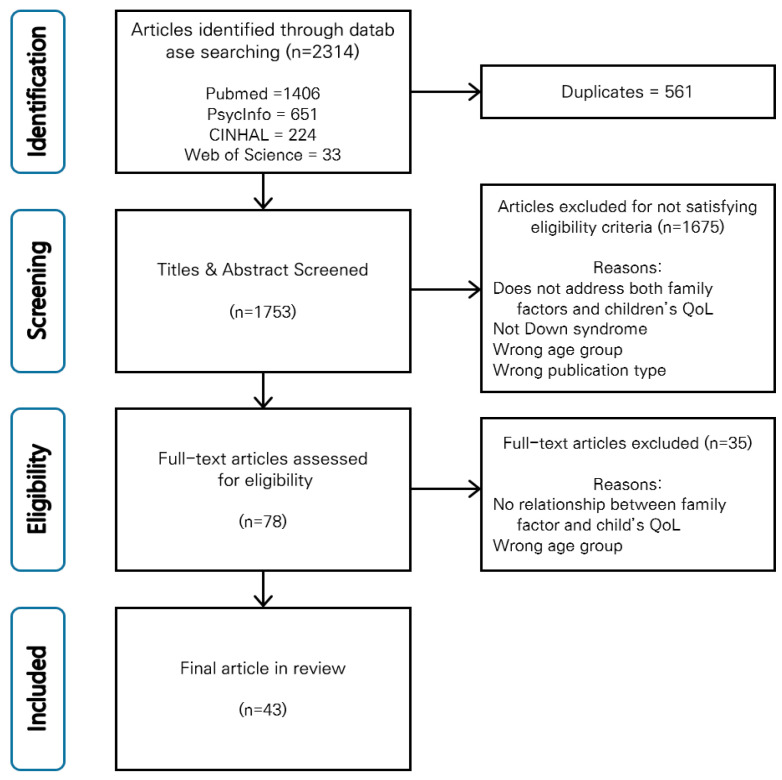
PRISMA flow chart of the search process.

**Table 1 ijerph-18-00419-t001:** Core quality of life domains and commonly used indicators. Adapted from Table 1 of [8]. Reproduced with permission of AAIDD’s Copyright Clearance Center.

Domain		Indicators
Well-Being	Emotional well-being	Contentment (satisfaction, moods, enjoyment) Self-concept (identify, self-worth, self-esteem) Lack of stress (predictability and control)
	Physical well-being	Activities of daily living (self-care, mobility) Health (functioning, symptoms, fitness, nutrition) Leisure (recreation, hobbies)
	Material well-being	Financial status (income, benefits) Employment (work status, work environment) Housing (type of residence, ownership)
Independence	Personal development	Education (achievements, education status) Personal competence (cognitive, social, practical) Performance (success, achievement, productivity)
	Self-determination	Autonomy/personal control (independence) Goals and personal values (desires, expectations) Choices (opportunities, options, preferences)
Social Participation	Interpersonal relation	Interactions (social networks, social contacts) Relationships (family, friends, peers) Supports (emotional, physical, financial)
	Social inclusion	Community integration and participation Community roles (contributor, volunteer) Social supports (support networks, services)
	Rights	Human (respect, dignity, equality) Legal (citizenship, access, due process)

**Table 2 ijerph-18-00419-t002:** Family variables [21].

Family Variables	Definition
Family Demands	Stresses and strains on or in the family system created by critical family situations
Family Appraisal	Family members’ views on stressors as well as the family’s ability to deal with family demands
Family Resources	Strengths and capabilities of individual family members, the family unit, and the community
Family Problem-Solving and Coping	The process of acquiring, allocating, and using resources for managing family demands, which refers both to individual family members and the family system as a whole

**Table 3 ijerph-18-00419-t003:** Characteristics of the studies review.

Author(s)/Year/Country	Purpose	Design	Sample (Used for Analysis)	
			Family	Children
Adamson et al., (2015), United States [27]	To document how parents weave new words into on-going interactions with children who are just beginning to speak	Longitudinal, Quantitative	104 families; 104 parents	DS: 28, ASD: 23, TD: 53 (m = 24.9 months, % male = 62)
Alesi and Pepi, (2017), Italy [28]	To explore parental beliefs concerning involvement, facilitators/barriers and benefits of PA in young people with DS	Cross-sectional, Qualitative	13 families; 7 mothers, 6 fathers	DS: 13 (m = 17.4 years, % male = 70)
Al-Sufyani et al., (2014), Yemen [29]	To assess the oral hygiene and gingival health status of children with DS attending special needs schools in Sana’a, and also to determine the association between these outcomes and socio-demographic and clinical variables	Cross-sectional, Quantitative	101 families: 101 parents	DS: 101 (m = 10.5 years, % male = 67)
Alwhaibi and Aldugahishem, (2018), Saudi Arabia [50]	To explore factors affecting participation in PA in Saudi children with DS, from their mothers’ perspectives	Cross-sectional, Qualitative	36 families; 36 mothers	DS: 36 (m = 8.9 years, % male = 61)
Barr and Shields, (2011), Australia [30]	To explore the barriers and facilitators to PA for people with DS	Cross-sectional, Qualitative	18 families; 16 mothers, 4 fathers	DS: 18 (m = 9.9 years, % male = 38)
Bentenuto et al., (2016), Italy [51]	To analyze children’s exploratory and symbolic play according to the different levels of its complexity, compare maternal play in the three groups during play in terms of mothers’ demonstrations and solicitations of the play, and analyze the associations between maternal demonstrations and solicitations and children’s play	Cross-sectional, Quantitative	75 families; 75 mothers	DS: 25, ASD: 25, TD: 25 (m = 33.3 months)
Brown and Woods, (2016), United States [31]	To examine the frequencies and proportions of parent coaching strategies used by interventionists across routine contexts in intervention sessions, which coaching strategies are more likely to support parents’ contingent use of intervention strategies, and which intervention strategies are more likely to support children’s contingent use of communication acts	Cross-sectional, Quantitative	9 families; 9 parents	DS: 3, ASD: 3, DD: 3 (12–28 months, % male = 33)
Burke and Hodapp, (2014), United States [52]	To examine which children, parent, and parent–school characteristics correlated with maternal stress	Cross-sectional, Quantitative	965 families; 965 mothers	DS: 88, DD: 877 (m = 10.9 years, % male = 70)
Burke et al., (2012), United States [32]	To examine how the behaviors of individuals with DS relate to parent functioning during the adolescent years	Cross-sectional, Quantitative	42 families; 42 parents	DS: 42 (m = 15.1 years, % male = 62)
Channell et al., (2014), United States [53]	To examine longitudinal change in performance on the Brief IQ subtests of the Leiter-R across four annual time points in adolescents with DS, and the relationship of maternal IQ to performance and growth in performance on the Leiter-R	Longitudinal, Quantitative	20 families; 20 mothers	DS: 20 (m = 12.8 years, % male = 100)
Choi and Yoo, (2015), Korea [33]	To identify the factors related to resilience of the families of children with DS	Cross-sectional, Quantitative	126 families; 126 parents	DS: 126 (m = 5.4 years, % male = 56)
De Falco et al., (2008), Italy [66]	To focus on paternal contributions to children’s play in association with the effective quality of father–children interactions	Cross-sectional, Quantitative	19 families; 19 fathers	DS: 19 (m = 35.3 months, % male = 63)
De Falco et al., (2010), Italy [34]	To study how DS children’s play differs in solitary and in collaborative situations with mother or father, how maternal and paternal play with their DS children differs in collaborative play situations, if paternal and maternal play behaviors are associated to one another, and if there are associations between children’s and each parent’s play during collaborative play	Cross-sectional, Quantitative	20 families; 20 mothers, 20 fathers	DS: 20 (m = 36.1 months, % male = 65)
Deckers et al., (2017), Netherlands [35]	To investigate the receptive and expressive vocabulary development over a period of 1.6 years in children with DS, taking into account the predictive role of children- and environment-related predictors as found in TD children	Longitudinal, Quantitative	36 families; 36 parents	DS: 36 (m = 4.5 years, % male = 56)
Dimitrova et al., (2016), United States [36]	To ask whether parents of children with autism and parents of children with DS were as likely as parents of TD children to translate into words their children’s gestures that uniquely identified objects and whether such parental translations had the same facilitative effect on the vocabulary development of children with autism and with DS as it did for TD children	Longitudinal, Quantitative	66 families; 66 parents	DS: 23, ASD: 20, TD: 23 (m = 27.5 months, % male = 83)
Dolva et al., (2014), Norway [37]	To describe the actual leisure activities of Norwegian adolescents with DS and to explore the influences on this participation	Cross-sectional, Mixed methods	38 families; 34 mothers, 4 fathers	DS: 38 (m = 14 years, % male = 50)
Downs et al., (2013), United Kingdom [38]	To explore PA of children and young people with DS from birth, specifically exploring the opportunities available to young people with DS and perceived barriers to physical activities	Cross-sectional, Qualitative	8 families; 8 parents	DS: 8 (m = 16.4 years, % male = 38)
Estigarribia et al., (2012), United States [54]	To examine which cognitive, environmental, and speech/language variables predict expressive syntax in boys with FXS, DS, and TD, and whether predictive relationships differed by group	Cross-sectional, Quantitative	90 families; 90 mothers	DS: 27, FXS: 38, TD: 25 (m = 8.8 years, % male = 100)
Evans and Uljarević, (2018), United States [39]	To examine the role of parental education and its possible influence on the cognitive ability (IQ) in propends with DS	Longitudinal, Quantitative	43 families; 43 parents	DS: 43 (m = 11.7 years, % male = 42)
Gilmore et al., (2009), Australia [55]	To examine maternal behaviors and their relationships with children’s mastery behaviors in a group of children with DS and a group of TD children of the same mental age	Cross-sectional, Quantitative	68 families; 68 mothers	DS: 25, TD: 43 (m = 42.9 months, % male = 56)
Howell et al., (2007), United States [56]	To examine characteristics of the children at age 3 as well as family income and emotional climate as predictors of children’s reported feelings of loneliness at school during middle childhood (age 10)	Longitudinal, Quantitative	82 families; 82 mothers	DS: 26, MI: 26, DD: 30 (m = 3 years, % male = 52)
Hung et al., (2011), Taiwan [68]	To describe the hospitalization profiles which include medical expenses and length of stays, and to determine their possible influencing factors of hospital admission on persons with DS in Taiwan	Cross-sectional, Quantitative	375 families	DS: 375 (m = 16.8 years, % male = 45)
Izquierdo-Gomez et al., (2015), Spain [40]	To identify potential correlates of sedentary time and television viewing time in youth with DS	Cross-sectional, Quantitative	98 families; 98 parents	DS: 98 (m = 15.3 years, % male = 64)
Karaaslan and Mahoney, (2013), Turkey [57]	To evaluate Responsive Teaching with a sample of 15 Turkish preschool-aged children with DS and their mothers over a six-month period of time	Intervention, RCT	15 families; 15 mothers	DS: 15 (m = 49.3 months, % male = 33)
Karaaslan, (2016), Turkey [41]	To compare how children with DS or autism interact with their mothers and their fathers, compare mothers’ and fathers’ style of interacting with their children, and determine whether there are differences in the interactive characteristics of mothers and fathers associated with the level of engagement exhibited by children with autism versus those with DS	Cross-sectional, Quantitative	27 families; 27 mothers, 27 fathers	DS: 11, ASD: 16 (m = 56.3 months, % male = 63)
Lyons et al., (2016), Ireland [42]	To explore parental views of their children’s participation and, identify barriers and facilitators in relation to participation in everyday activities	Cross-sectional, Qualitative	5 families; 7 parents	DS: 5 (m = 8.8 years, % male = 40)
Malmir et al., (2015), Iran [58]	To assess the relation between mother’s happiness level with cognitive-executive functions and facial emotional recognition ability as two factors in learning and adjustment skills in mentally retarded children with DS	Cross-sectional, Quantitative	30 families; 30 mothers	DS: 30 (m = 10.5 years, % male = 70)
Minnes et al., (2015), Canada [59]	To explore predictors of both parent perceived positive gain and parent distress, reported by parents whose young children with DD were in the process of transitioning into school	Cross-sectional, Quantitative	155 families; 155 mothers	DS: 19, ASD: 81, UID/DD: 40, OGC: 6, OD: 9 (m = 4.9 years, % male = 73)
Mitchell et al., (2015), United States [60]	To extend the investigation of possible differences in dimensions of parenting stress and also examine whether differences exist in maternal and children contingent responsiveness during mother–children interaction in two groups (DS and UDD)	Cross-sectional, Quantitative	97 families; 97 mothers	DS: 43, UDD: 54 (m = 3 years, % male = 51)
Niccols et al., (2011), Canada [61]	To examine maternal sensitivity in mothers of children with Down syndrome at age 2, 3, and 5 years, and relations with physical and verbal aggression at home and school at age 5	Longitudinal, Quantitative	53 families; 53 mothers	DS: 53 (m = 19 months, % male = 47)
Oates et al., (2011), Australia [43]	To investigate how for children with DS the International Classification of Functioning, Disability and Health components of impairment in body functions or structures, as well as personal and environmental factors related to their participation in friendships and leisure	Cross-sectional, Quantitative	208 families; 208 parents	DS: 208 (5–18 years, % male = 57)
Phillips et al., (2016), United States [62]	To compare the parenting styles and dimensions in mothers of children with DS and mothers of TD children	Cross-sectional, Quantitative	82 families; 82 mothers	DS: 35, TD: 47 (m = 8.5 years, % male = 54)
Polfuss et al., (2017), United States [44]	To explore associations among parental feeding behaviors, parent weight concerns, demographics, and children weight status in a sample of 356 parents of children diagnosed with ASD, SB, and DS	Cross-sectional, Quantitative	356 families; 356 parents	DS: 110, SB: 99, ASD: 147 (m = 2.5–19.5 years, % male = 66)
Porto-Cunha and Limongi, (2010), Brazil [45]	To verify the influence of environmental and contextual variables in the pragmatic aspects of language of DS children when interacting with their caregivers and therapist, and to compare their performance in both situations	Cross-sectional, Quantitative	15 families; 15 caregivers	DS: 15 (m = 4–6 years)
Scott et al., (2014), Australia [26]	To explore what makes for a ‘‘good life’’ from the perspective of young adults with DS and, Identify the barriers and facilitators to participation	Cross-sectional, Qualitative	12 families	DS: 12 (m = 21 years, % male = 50)
Terrone et al., (2014), Italy [67]	To investigate whether family relationships more oriented towards recognizing the maturational processes of adolescents with DS are positively related to the construction of an adequate self-representation, and developmental paths of personal and social autonomy and adaptive behaviors	Cross-sectional, Quantitative	170 families	DS: 85, TD: 85 (m = 20.7 years, % male = 57)
Turner et al., (2008), United Kingdom [46]	To identify the contemporary and antecedent predictors of the level of academic attainment achieved by a representative sample of young people with DS	Longitudinal, Quantitative	71 families	DS: 71 (m = 9 years, % male = 58)
Venuti et al., (2008), Italy [63]	To investigate mother–children interaction and its associations with play in children with DS	Cross-sectional, Quantitative	28 families; 28 mothers	DS: 28 (m = 35.6 months, % male = 68)
Venuti et al., (2009), Italy [64]	To compare the structure of play in the two groups, the effects of mothers’ participation on children’s play in the two groups, children with DS and TD children for their relative order between solitary to collaborative play situations, maternal play in the two groups, and children and mothers in the two groups in terms of their attunement and synchrony during play	Cross-sectional, Quantitative	54 families; 54 mothers	DS: 21, TD: 33 (m = 25.8 months)
Venuti et al., (2012), Italy [65]	To compare maternal functional language directed to children with two DD (ASD and DS) with TD children and to investigate relations of maternal functional language with children language skills	Cross-sectional, Quantitative	60 families; 60 mothers	DS: 20, ASD: 20, TD: 20 (m = 39.6 months)
Wang et al., (2007), China [47]	To evaluate social adjustment and related factors among Chinese children with DS	Cross-sectional, Quantitative	106 families; 106 parents	DS: 36, TD: 70 (m = 81.1 months, % male = 54)
Wasant et al., (2008), Thailand [48]	To analyze the factors including both children and family factors that influence development in the first three years of DS children	Cross-sectional, Quantitative	100 families	DS: 100 (3–6 years, % male = 59)
Westerveld and Van Bysterveldt, (2017), Australia, New Zealand [49]	To investigate if there were differences in the home literacy environments of preschool children on the autism spectrum and preschool children with DS to determine if the home literacy environment may potentially be associated with strengths or weaknesses in children’s social communication skills	Cross-sectional, Quantitative	111 families; 111 parents	DS: 31, ASD: 80 (m = 54.8 months, % male = 75)

Note. ASD = autism spectrum disorder, DD = developmental delay, DS = Down syndrome, FXS = fragile X syndrome, IQ = intelligence quotient, M = mean age, MI = motor impairments, OD = other diagnosis, OGC = other genetic conditions, PA = physical activity, SB = spina bifida, TD = typically developing, UDD = undifferentiated developmental disabilities, UID = unspecified intellectual disability.

**Table 4 ijerph-18-00419-t004:** Family focuses in stated aims of reviewed studies.

Family Focus	Studies
**Fully focused on family–child quality of life relationships** The primary aim of the study was to examine the relationship between family factors reflected in the Resiliency Framework (family demands, family appraisal, family resources, and family problem-solving and coping) and child quality of life. If there were multiple aims all addressed these relationships.	Adamson et al., 2015 [27]
	Burke et al., 2012 [32]
	Choi and Yoo, 2015 [33]
	De Falco et al., 2008 [66]
	De Falco et al., 2010 [34]
	Deckers et al., 2017 [35]
	Dimitrova et al., 2016 [36]
	Evans and Uljarević, 2018 [39]
	Gilmore et al., 2009 [55]
	Howell et al., 2007 [56]
	Karaaslan and Mahoney, 2013 [57]
	Karaaslan, 2016 [41]
	Malmir et al., 2015 [58]
	Minnes et al., 2015 [59]
	Mitchell et al., 2015 [60]
	Niccols et al., 2011 [61]
	Oates et al., 2011 [43]
	Phillips et al., 2017 [62]
	Polfuss et al., 2017 [44]
	Porto-Cunha and Limongi, 2010 [45]
	Terrone et al., 2014 [67]
	Turner et al., 2008 [46]
	Venuti et al., 2008 [63]
	Venuti et al., 2009 [64]
	Venuti et al., 2012 [65]
	Westerveld and Van Bysterveldt, 2017 [49]
**Partially focused on family–child quality of life relationships** The relationship between family factors as defined by the Resiliency Framework and child quality of life was one of several research aims.	Al-Sufyani et al., 2014 [29]
	Bentenuto et al., 2016 [51]
	Brown and Woods, 2016 [31]
	Burke and Hodapp, 2014 [52]
	Channell et al., 2014 [53]
	Estigarribia et al., 2012 [54]
	Izquierdo-Gomez et al., 2015 [40]
	Lyons et al., 2016 [42]
	Wang et al., 2007 [47]
	Wasant et al., 2008 [48]
**Minimally focused on family–child quality of life relationships** The relationship between family factors as defined by the Resiliency Framework and child quality of life was not reflected in the study aims was assessed and reported in the analysis.	Alesi and Pepi, 2017 [28]
	Alwhaibi and Aldugahishem, 2018 [50]
	Barr and Shields, 2011 [30]
	Dolva et al., 2014 [37]
	Downs et al., 2013 [38]
	Hung et al., 2011 [68]
	Scott et al., 2014 [26]

**Table 5 ijerph-18-00419-t005:** Summary of family variables related to children’s quality of life variable.

Child QoL Variables		Family Variables				
		Family Demands	Family Appraisal	Family Resources		Family Problem-Solving and Coping
Well-Being	Overall Well-being	Scott et al., 2014 [26]		Scott et al., 2014 [26]		
	Emotional Well-being	Scott et al., 2014 [26]		Howell et al., 2007 [56] Scott et al., 2014 [26]		
	Physical Well-being	Alesi and Pepi, 2017 [28] Alwhaibi and Aldugahishem, 2018 [50] Barr and Shields, 2011 [30] Choi and Yoo, 2015 [33] Downs et al., 2013 [38] Izquierdo-Gomez et al., 2015 [40] Polfuss et al., 2017 [44]	Izquierdo-Gomez et al., 2015 [40]	Alesi and Pepi, 2017 [28] Al-Sufyani et al., 2014 [29] Alwhaibi and Aldugahishem, 2018 [50] Barr and Shields, 2011 [30] Choi and Yoo, 2015 [33] Dolva et al., 2014 [37] Downs et al., 2013 [38] Hung et al., 2011 [68] Izquierdo-Gomez et al., 2015 [40]	Oates et al., 2011 [43] Polfuss et al., 2017 [44] Wang et al., 2007 [47]	Choi and Yoo, 2015 [33] Izquierdo-Gomez et al., 2015 [40] Polfuss et al., 2017 [44]
	Material Well-being	Scott et al., 2014 [26]		Scott et al., 2014 [26]		
Independence	Personal Development	Burke and Hodapp, 2014 [52] Burke et al., 2012 [32] Choi and Yoo, 2015 [33] Minnes et al., 2015 [59] Mitchell et al., 2015 [60] Phillips et al., 2017 [62] Turner et al., 2008 [46]	Burke et al., 2012 [32] Minnes et al., 2015 [59]	Channell et al., 2014 [53] Choi and Yoo, 2015 [33] de Falco et al., 2008 [66] Deckers et al., 2017 [35] Estigarribia et al., 2012 [54] Evans and Uljarević, 2018 [39] Howell et al., 2007 [56] Karaaslan and Mahoney, 2013 [57] Karaaslan, 2016 [41] Malmir et al., 2015 [58] Minnes et al., 2015 [59]	Mitchell et al., 2015 [60] Niccols et al., 2011 [61] Phillips et al., 2017 [62] Porto-Cunha and Limongi, 2010 [45] Terrone et al., 2014 [67] Turner et al., 2008 [46] Venuti et al., 2008 [63] Wang et al., 2007 [47] Wasant et al., 2008 [48]	Adamson et al., 2015 [27] Bentenuto et al., 2016 [51] Brown and Woods, 2016 [31] Burke et al., 2012 [32] Choi and Yoo, 2015 [33] de Falco et al., 2008 [66] de Falco et al., 2010 [34] Dimitrova et al., 2016 [36] Gilmore et al., 2009 [55] Minnes et al., 2015 [59] Turner et al., 2008 [46] Venuti et al., 2009 [64] Venuti et al., 2012 [65] Westerveld and Van Bysterveldt, 2017 [49]
	Self-Determination	Burke et al., 2012 [32]	Burke et al., 2012 [32]	Wang et al., 2007 [47]		Burke et al., 2012 [32]
Social Participation	Interpersonal Relation			Oates et al., 2011 [43]		
	Social Inclusion	Lyons et al., 2016 [42]		Lyons et al., 2016 [42]		
	Rights	Choi and Yoo, 2015 [33]		Choi and Yoo, 2015 [33]		Choi and Yoo, 2015 [33]

## Data Availability

Not applicable.

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
