# Peer review of "Family Variables and Quality of Life in Children with Down Syndrome: A Scoping Review"

_ijerph, 2021, doi:10.3390/ijerph18020419_

Round 1

Reviewer 1 Report

This review focuses on the inter-relations between family variables and quality of life in children with Down syndrome. The review is methodologically sound and will make a nice contribution to the field.

My suggestions mainly refer to the '3.2 Family Variables and Children's QoL' and '3.3 Degree of Family Focus' sections. As they are, the particular sections repeat the information of the review's Tables, while they would be much more informative if the main findings of the studies included were mentioned. I realize that the studies are many, yet, the authors could highlight in both sections the most important findings concerning the contribution of family variables to the quality of life of children with Down syndrome.

Furthermore, have the authors noticed a modification/change in the inter-relation between family variables and children's QoL in different age-groups across the studies?

In the same vein, has the children's language, cognitive and social functioning phenotype had a role in the relations examined? I realize that such questions do not exactly fall within the scope of the review, but they could add an interesting dimension to the variables already examined and discussed. The authors touch upon this point in line 356 in the Discussion, but I believe that the particular issue could have been much more elaborated upon in the review.

Regarding Discussion, lines 327 till 346 do not add an interpretive dimension to the review, as Discussion sections should do. The particular part lacks a critical approach to the studies being reviewed, so they should be modified accordingly.

In the last paragraph of the Discussion, the authors set as future research goals the investigation of the relationship between family variables and quality of life in children with Down Syndrome cross-culturally. In lines 150-155, I noticed that a non-negligible portion of the studies included in the review were conducted outside the USA or Europe (e.g. in Turkey, China, Iran, Brazil, Korea, Saudi Arabia, Taiwan, Thailand, Yemen). Were findings from the particular studies differentiated in any possible way(s) from the studies conducted in the USA and Europe, and if they were, which are the reasons underlying possible differences?

Minor points

line 107: was used (add 'to') screen

line 201: No relationship between family factor and...(is there something missing in the figure?)

lines 347-354: references should be added

lines 365-368: references should be added

line 415: and their contribution (replace 'it')

Author Response

Response to Reviewer 1 Comments

We greatly appreciate your thoughtful comments, which helped us improve our manuscript. Please refer to our responses to your comments as follows.

Point 1: This review focuses on the inter-relations between family variables and quality of life in children with Down syndrome. The review is methodologically sound and will make a nice contribution to the field.

My suggestions mainly refer to the '3.2 Family Variables and Children's QoL' and '3.3 Degree of Family Focus' sections. As they are, the particular sections repeat the information of the review's Tables, while they would be much more informative if the main findings of the studies included were mentioned. I realize that the studies are many, yet, the authors could highlight in both sections the most important findings concerning the contribution of family variables to the quality of life of children with Down syndrome.

Response 1: Because this is a scoping review, our focus was on identifying areas where the volume of research was such that an integrative or systematic review could be undertaken. We also wanted to highlight potentially important areas of inquiry where little work had been completed. In the discussion section, we highlighted the implications of trends and gaps we identified for future research. Based on the results of our review, we made specific recommendations for further research syntheses and future research.

In terms of the 3.3 Degree of Family Focus section, we also focused in identifying areas of research emphasis and gaps. We were especially interested in examining the extent to which family variables were a primary or secondary focus on the research. We concluded that that most of the reviewed studies focused on individual family members although most of them were categorized as fully or partially family-focused studies. Thus, we pointed out the need for additional studies including family system measures considering the family as a whole.

Additionally, we have added the results and interpretation regarding the new analyses that if there were significant relationships between family variables identified by family system measures and child QoL variables. We have found that all nine studies but one showed significant relationships, which addresses the contribution of family systems variables to child QoL. We have revised the manuscript including this information (Lines 313 - 319, 408 - 411).

Point 2: Furthermore, have the authors noticed a modification/change in the inter-relation between family variables and children's QoL in different age-groups across the studies? In the same vein, has the children's language, cognitive and social functioning phenotype had a role in the relations examined? I realize that such questions do not exactly fall within the scope of the review, but they could add an interesting dimension to the variables already examined and discussed. The authors touch upon this point in line 356 in the Discussion, but I believe that the particular issue could have been much more elaborated upon in the review.

Response 2: We agree that the relationships between family variables and QoL in children with DS could vary according to children’s characteristics such as children’s age, language, and social functioning. However, the examination of these relationships was beyond the aims of the current scoping review. Although beyond the scope of the current review to examine the relationship between characteristics of the child, we have added additional information on the study sample (children’s age; % male) to Table 3.

Point 3: Regarding Discussion, lines 327 till 346 do not add an interpretive dimension to the review, as Discussion sections should do. The particular part lacks a critical approach to the studies being reviewed, so they should be modified accordingly.

Response 3: The parts you pointed out were to summarize the main results of the current scoping review; the reviewed studies particularly focused on relationships between family resources variables and QoL variables regarding personal development and physical well-being of children with DS. After the summarization, the next paragraphs provide our interpretation of research trends and gaps. Although these are not interpretive, the first paragraphs of the discussion are intended to set the stage for the interpretive discussion that follows.

Point 4: In the last paragraph of the Discussion, the authors set as future research goals the investigation of the relationship between family variables and quality of life in children with Down Syndrome cross-culturally. In lines 150-155, I noticed that a non-negligible portion of the studies included in the review were conducted outside the USA or Europe (e.g. in Turkey, China, Iran, Brazil, Korea, Saudi Arabia, Taiwan, Thailand, Yemen). Were findings from the particular studies differentiated in any possible way(s) from the studies conducted in the USA and Europe, and if they were, which are the reasons underlying possible differences?

Response 4: The current analysis did not examine the differences of findings according to different countries since this scoping review purported to identify the extent and nature of the research filed regarding relationships family and QoL in children with DS.

However, prior studies have indicated that the perception or belief on disability could be different by different cultures. In addition, the medical and societal system for individuals with a disability could vary across countries. Accordingly, we advanced recommendations for future research to examine possible differences in relationships between family and child QoL variables according to different cultures. We have added more explanations about this in the discussion section (Lines 433 - 438).

Minor points

Authors’ responses

line 107: was used (add 'to') screen

We have revised it.

line 201: No relationship between family factor and...(is there something missing in the figure?)

We have revised the boxes to show the whole text in the figure.

lines 347-354: references should be added

We have added references.

lines 365-368: references should be added

We have added references.

line 415: and their contribution (replace 'it')

We have revised as follows.

Particularly, family resources and their contribution to the personal development of children with DS could be targeted for systematic reviews in that the current review found evidence of a sufficient number of relevant studies.

Reviewer 2 Report

This scoping review about family variables and quality of life in children with down syndrome addresses an important and timely topic. Considered authors’ specific aims, the methodological choice to implement a scoping review is adequate and it was rigorously performed. I appreciated also the choice to use of the resiliency model and the Schalock’s conceptualization of children’s QoL to categorize included studies. I have few suggestions for improvement.

- In the introduction section authors well justified the choice of the two theoretical models, however they could deepen the description of the state of the art about findings of quality of life in individuals with DS. Authors reported that findings provide a mixed picture and I think that a brief explanation is needed.

- In the introduction section (lines 50-52), authors reported as a limit of the previous studies the scarce consideration of individual variables that may act as moderator or influence the QoL, however mapping the selected records they considered only children’s variables linked with QoL. Eventual other individual variables tested by the selected studies may be added in tables.

- Authors may better explicit the choice of the year range (2007-2018) identified for the literature search.

- I agree with the choice of the children’s age range, however it’s a wide range (0-21) and different family variables may be salient in different developmental stages. I think that children’ age (mean and SD and/or range) may be added in Table 4 for each study.

- The order of some tables and contents may be modified in order to make the manuscript easier to read: e.g. tables 3 and 4 with the description of selected studies precede the description of study selection process (and Figure 1). Table 3 is presented at pages 4-5, but that part is described in text some pages later (line 290 – section “degree of family focus”).

- Please include also a brief section about limitations of the scoping review process

Author Response

Response to Reviewer 2 Comments

We greatly appreciate your thoughtful comments, which helped us improve our manuscript. Please refer to our responses to your comments as follows.

Point 1: This scoping review about family variables and quality of life in children with down syndrome addresses an important and timely topic. Considered authors’ specific aims, the methodological choice to implement a scoping review is adequate and it was rigorously performed. I appreciated also the choice to use of the resiliency model and the Schalock’s conceptualization of children’s QoL to categorize included studies. I have few suggestions for improvement.

In the introduction section authors well justified the choice of the two theoretical models, however they could deepen the description of the state of the art about findings of quality of life in individuals with DS. Authors reported that findings provide a mixed picture and I think that a brief explanation is needed.

Response 1: We have updated the introduction to include results from recent empirical studies examining QoL of children and adolescents with DS (Lines 49 - 61).

Point 2: In the introduction section (lines 50-52), authors reported as a limit of the previous studies the scarce consideration of individual variables that may act as moderator or influence the QoL, however mapping the selected records they considered only children’s variables linked with QoL. Eventual other individual variables tested by the selected studies may be added in tables.

Response 2: The current scoping review focused on identifying a child’s QoL variables based on Schalock’s conceptualization that had been studied in relation to family variables. The examination of other variables more than the family and child QoL variables would be beyond the aims of the current review.

Point 3: Authors may better explicit the choice of the year range (2007-2018) identified for the literature search.

Response 3: We have added more explanation about the choice of the year range in the method section (Lines 108 -110).

Point 4: I agree with the choice of the children’s age range, however it’s a wide range (0-21) and different family variables may be salient in different developmental stages. I think that children’ age (mean and SD and/or range) may be added in Table 4 for each study.

Response 4: We have added children’s age of each study in Table 3 and the relevant information in the result section (Lines165 - 166).

Point 5: The order of some tables and contents may be modified in order to make the manuscript easier to read: e.g. tables 3 and 4 with the description of selected studies precede the description of study selection process (and Figure 1). Table 3 is presented at pages 4-5, but that part is described in text some pages later (line 290 – section “degree of family focus”).

Response 5: We have reordered the figure and tables to place the tables very near the relevant description in the text. As a result, the prior table 4 which describes the characteristics of reviewed studies has become Table 3. The prior table 3 which explains family focuses in stated aims of reviewed studies has become Table 4.

Point 6: Please include also a brief section about limitations of the scoping review process.

Response 6: We have added the limitations of this study in the discussion section (Lines 441 - 447).

Reviewer 3 Report

The manuscript describes high-level work. The scoping review is highly informative and interesting to guide future research on this topic.

Despite not proposing any major changes, I recommend considering the following: the manuscript focuses on individuals with Down Syndrome aged 0-21 years. However, the title and keywords refer primarily to "children." Although the authors justify this use based on the United Nations Convention on the Rights of the Child (children are people up to the age of 18), the term can be somewhat misleading for readers as the review deals with individuals up to 21 years, and includes what are commonly referred to as "children and adolescents."

I recommend introducing the term "adolescents", at least in the keywords and in those parts of the text where it is necessary.

In this sense, despite the fact that the tables and descriptions are very complete, the information on the samples of the different studies reviewed does not indicate at any time the ages of the individuals with DS (or other groups studied). That information would be relevant and should be included in Table 4 and related texts (description or discussion). The absence of this information is difficult to understand, given the high level of the work.

Additionally, it would also be relevant to know how sex is distributed in the study samples. It is considered necessary to include this information in the table, as well as a reference and discussion in the text.

Author Response

Response to Reviewer 3 Comments

We greatly appreciate your thoughtful comments, which helped us improve our manuscript. Please refer to our responses to your comments as follows.

Point 1: The manuscript describes high-level work. The scoping review is highly informative and interesting to guide future research on this topic.

Despite not proposing any major changes, I recommend considering the following: the manuscript focuses on individuals with Down Syndrome aged 0-21 years. However, the title and keywords refer primarily to "children." Although the authors justify this use based on the United Nations Convention on the Rights of the Child (children are people up to the age of 18), the term can be somewhat misleading for readers as the review deals with individuals up to 21 years, and includes what are commonly referred to as "children and adolescents." I recommend introducing the term "adolescents", at least in the keywords and in those parts of the text where it is necessary.

Response 1: We have added ‘adolescents’ as keywords. Also, in lines 36 and 57, we have added ‘adolescents.’

Point 2: In this sense, despite the fact that the tables and descriptions are very complete, the information on the samples of the different studies reviewed does not indicate at any time the ages of the individuals with DS (or other groups studied). That information would be relevant and should be included in Table 4 and related texts (description or discussion). The absence of this information is difficult to understand, given the high level of the work.

Response 2: We have added the information regarding children’s age of reviewed studies in Table 3 and the result section (Lines 165 -166).

Point 3: Additionally, it would also be relevant to know how sex is distributed in the study samples. It is considered necessary to include this information in the table, as well as a reference and discussion in the text.

Response 3: We have added the information regarding the sex ratio of the children of reviewed studies in Table 3 as well as the result section (Lines 165 -166).

Round 2

Reviewer 1 Report

I am satisfied with the revisions made. The authors have addressed all my concerns in the revised manuscript. I recommend acceptance in present form.